# Epigenetic Mechanisms in Hepatic Stellate Cell Activation During Liver Fibrosis and Carcinogenesis

**DOI:** 10.3390/ijms20102507

**Published:** 2019-05-21

**Authors:** Marina Barcena-Varela, Leticia Colyn, Maite G. Fernandez-Barrena

**Affiliations:** 1Hepatology Program, CIMA, University of Navarra, 31180 Pamplona, Spain; mbarcena.1@alumni.unav.es (M.B.-V.); lcolyn@alumni.unav.es (L.C.); 2CIBERehd, Instituto de Salud Carlos III, 28029 Madrid, Spain; 3Instituto de Investigaciones Sanitarias de Navarra-IdiSNA, 31180 Pamplona, Spain

**Keywords:** Tumor microenvironment, fibrosis, hepatic stellate cells, epigenetic remodeling

## Abstract

Liver fibrosis is an essential component of chronic liver disease (CLD) and hepatocarcinogenesis. The fibrotic stroma is a consequence of sustained liver damage combined with exacerbated extracellular matrix (ECM) accumulation. In this context, activation of hepatic stellate cells (HSCs) plays a key role in both initiation and perpetuation of fibrogenesis. These cells suffer profound remodeling of gene expression in this process. This review is focused on the epigenetic alterations participating in the transdifferentiation of HSCs from the quiescent to activated state. Recent advances in the field of DNA methylation and post-translational modifications (PTM) of histones (acetylation and methylation) patterns are discussed here, together with altered expression and activity of epigenetic remodelers. We also consider recent advances in translational approaches, including the use of epigenetic marks as biomarkers and the promising antifibrotic properties of epigenetic drugs that are currently being used in patients.

## 1. Introduction

Liver cancer is a major health problem worldwide causing more than 700,000 deaths annually [1]. Hepatocellular carcinoma (HCC) development is strictly dependent on environmental cues able to induce and maintain biological changes in tumor cells and in their tissue niche. Over the years, the tumor microenvironment has evolved as an important target to understand tumor progression, clinical prognosis, and treatment responses in cancer [2,3]. Understanding the dynamics of surrounding nontumor cells and noncellular components secreted from the microenvironment will allow us to better understand the mechanisms involved in hepatocarcinogenesis and the design of more effective therapeutic approaches.

HCC usually develops on the background of chronic liver disease (CLD), a pathological process that involves the progressive and sustained destruction and self-regeneration of the liver parenchyma leading to fibrosis and cirrhosis, and in the end stage, the development of cancer. CLD consists of a wide range of pathological processes which include persistent inflammation, a protracted wound-healing response and scar formation, and eventually malignant cellular transformation [4,5].

The link between hepatic stroma and cancer is not just a sequential process; it should be regarded as an interactive crosstalk between cellular and molecular players taking place along the development of CLD. Although most evidence suggests that fibrosis promotes HCC, it is possible that in some clinical settings fibrosis and HCC might occur due to the same underlying factors rather than one promoting the other [6]. A central event at the beginning of CLD after sustained damage, independent of the etiology, is hepatocellular death that triggers an inflammatory reaction [7]. This reaction is linked to a potent regenerative response in order to restore the lost hepatic tissue. In the context of acute or self-limited damage, this inflammatory and wound-healing response is transient, and the liver architecture is restored to its normal state. However, under chronic injury, the response is sustained, leading to the accumulation of extracellular matrix (ECM), a process known as hepatic fibrogenesis. If the situation is prolonged in time, it leads to the progressive substitution of liver parenchyma for scar tissue, resulting in a cirrhotic state where liver histology and function are greatly affected and the chances of developing cancer increases considerably [8,9,10].

There is a multitude of cellular and molecular components, such as cytokines, growth factors, and metabolites, undergoing complex interaction during fibrogenesis. However, hepatic stellate cell (HSC) activation remains the most dominant pathway to produce the ECM necessary to allow the restoration of damaged liver mass [7,8,11] and constitutes the leading cause of hepatic fibrosis perpetuation [12]. HSCs are differentiated mesenchymal cells present in the perisinusoidal space, which embrace endothelial cells and focally provide a double lining for the sinusoid [13]. The most characteristic feature of stellate cells in normal liver is their cytoplasmic storage of vitamin A (retinoid) droplets. Almost all the vitamin A in the liver is stored in stellate cells. Other functions of HSCs in normal liver include the control of extracellular matrix turnover, and the regulation of sinusoids contractility [13]. However, hepatocellular damage activates the transformation of quiescent stellate cells into myofibroblast-like cells that play a key role in the development of the inflammatory and fibrogenic responses. Therefore, following liver injury, HSCs become activated, acquiring a phenotype characterized by the loss of retinoid droplets, increased proliferation and contractile activity, and the release of proinflammatory, profibrogenic, and promitogenic cytokines [8,10,13]. How all these coordinated changes are globally orchestrated in these cells is still unknown. The transdifferentiation process results from either activating events as well as from the loss of repressive signaling [13]. HSCs undergo a drastic phenotypic change to become myofibroblasts [14], involving extensive changes in gene expression. However, these alterations are not necessarily irreversible. In fact, it was demonstrated that activated HSCs can return to the quiescent state upon resolution of chronic liver injury [15,16]. These findings suggest that epigenetic mechanisms might be crucial in this process, controlling the cellular plasticity of HSCs and fibrogenesis. 

The term “epigenetic” refers to hereditable and long-term changes in gene expression that do not necessarily involve mutations in DNA sequences. How the same genetic information is translated into different cellular identities is a process mainly regulated at the epigenetic level. Epigenetic regulators and transcription factors act to organize the genome into accessible or closed regions orchestrating the appropriate transcriptional program in any given cell. As such, epigenetic regulation is fundamental to maintain cellular identity, and the different physical and biological phenotypes [17]. 

The transdifferentiation of quiescent HSCs to a profibrogenic myofibroblastic phenotype requires global epigenetic remodeling to bring about the suppression of adipogenic differentiation factors, the de novo expression of regulators for the new phenotype and cell cycle entry [18]. Epigenetic events that control HSC activation and function are highly dynamic processes and the phenotypic conversion requires changes in expression of hundreds of different genes [19]. However, modulation of gene expression is not only important for generation of the HSCs’ activated state, but also for their ability to respond to cues from their immediate microenvironment and their persistence in CLD up to cancer development. In the present review, we summarize what is known about the most relevant epigenetic mechanisms controlling HSC transdifferentiation and hepatic fibrogenesis.

## 2. Changes in DNA Methylation during HSCs Activation

DNA methylation is the best understood and most extensively studied epigenetic mechanism. DNA methylation occurs in cytosine bases that are located 5’ to a guanosine base in a CpG dinucleotide. Methylation of CpGs in promoter regions is usually linked to the inactivation of gene expression, and thus is commonly considered as a repressive mark. Methylated CpG islands may promote condensation of chromatin that can directly inhibit the interaction of DNA binding proteins with their target sites and can also provide recognition signals for the recruitment of methyl-CpG-binding domain-containing proteins (i.e., MeCP2, MBD1-4, and Kaiso) along with their associated complexes [20]. Recent studies also suggest that depending on their location and density along the structure of genes, they might also be acting as activating marks [21,22], as methylation in gene bodies and at CpG-poor sites have been observed in active genes [23]. However, the mechanisms responsible for this remain poorly defined but are thought to be related to structural requirements for transcription elongation [21]. The enzymes responsible for DNA methylation are known as DNA methyltransferases (DNMTs) and are classified in two main categories: those involved in de novo DNA methylation (DNMT3A and DNMT3B) and those involved in the maintenance of postreplicative DNA methylation (DNMT1) [24,25,26]. Nevertheless, it has become apparent that depending on the DNA sequence context DNMT1 can display de novo activity, while DNMT3A and DNMT3B may participate in maintenance methylation [27,28]. Another mechanism affecting the levels of methyl groups present on CpG residues is the DNA demethylation process. The DNA demethylation process starts with the oxidation of 5-methylcytosine (5mC) to 5-hydroxymethylcytosine (5hmC), a reaction catalyzed by the family of 2-oxoglutarate-dependent dioxygenases such as the Ten Eleven Translocation (TET) enzyme family, including TET1, TET2, and TET3. These enzymes participate in the initiation of DNA demethylation as an active process [29,30]. 

Development of fibrosis has been associated with changes in DNA methylation patterns and in epigenetic enzymes expression and activity [9,31,32]. Mann´s group was a pioneer in investigating the role of DNA methylation in HSC activation. They carried out in vitro experiments where they tested the effects of the DNA methylation inhibitor 5-aza-2-deoxycytidine (Decitabine), observing that inhibition of DNA methylation blocks myofibroblastic differentiation of culture-activated primary rat HSCs [33]. Several studies have demonstrated the global hypomethylation of DNA in HSCs upon transdifferentiation or activation [34,35]. Cultured primary HSCs lose approximately 60% of the initial methylation level within the first three days of culture compared to quiescent cells [35]. Although HSC activation led to an overall loss of DNA methylation, further genome-wide DNA methylation analyses demonstrated that gene-specific DNA hypo- and hypermethylation at promoter and CpG-rich regions correlated with altered gene expression in activated cells [35,36]. These differentially methylated genes participate in relevant biological processes important for fibrogenesis such as “regulation of cell activation”, “immune response”, “response to wounding”, and “regulation of localization” [35]. Thus, DNA methylation changes control essential mechanisms of HSC activation. For example, the Wnt pathway, which participates in HSCs activation [37,38,39], has been found to be regulated by this epigenetic mechanism. *Apc2* and *Wnt5a*, two key players in Wnt signaling, increase their expression in association with DNA methylation changes during liver fibrosis as well as HSC activation [35,39]. Moreover, transcriptional activation of profibrogenic genes such as *Actg2*, *Loxl1*, *Loxl2*, and *Col4A1/2* has been correlated with a reduction in promoter methylation levels in activated HSCs [36]. Other demethylated and upregulated genes in HSCs are involved in the regulation of nucleotide metabolism, cell cycle progression, and signal transduction, consistent with the induction of proliferation upon activation of these cells. Conversely, transcriptional repression associated with DNA hypermethylation was found in genes such as *Adamts9* and *Mmp15* [36] that encode matrix metalloproteinases involved in ECM degradation, *Smad7* [40], an antagonist of TGFβR1 signaling, and *Pten* [41], whose loss of function has been associated with decreased cellular apoptosis. 

Interestingly, the changes in DNA methylation were found to be accompanied by changes in the expression of enzymes that regulate DNA methylation and hydroxymethylation. Indeed, there is a strong increase in the expression of DNMT3a and DNMT3b during rat HSC activation, while these enzymes are almost undetectable in freshly isolated HSCs [35]. DNMT1 is also upregulated during HSCs culture activation although the increase is not as drastic as that of DNMT3a and DNMT3b [35]. These results were validated in vivo in different animal models and in fibrotic liver tissues from patients. Furthermore, concomitant with increased DNMTs expression, it was found that TET enzymes tend to become downregulated in liver fibrosis and in activated HSCs [42]. This correlated with the loss of 5-hmC global levels in fibrotic livers, suggesting a role for this novel DNA modification in HSC activation and liver fibrogenesis [42]. 

## 3. Reprogramming of the Histone Code in HSCs Activation

Within the chromosomes, DNA is packaged into chromatin where the DNA coils around an octamer of histones comprising two H2A, two H2B, two H3, and two H4 subunits, forming the repeating unit, the nucleosome. Unstructured N-terminal tails of histones protruding out of the nucleosomes are targets for a variety of post-translational modifications (PTMs), including phosphorylation of serine residues, methylation of arginine or lysine residues, and acetylation, ubiquitination, sumoylation, and ADP-ribosylation of lysine residues. Histone modifications are highly variable and dynamic and conform to the so called “histone code”. Specific patterns of histone modifications, i.e. specific “codes”, have been involved in the regulation of gene expression [17,43]. Histone modifications influence the degree of chromatin compaction and modulate the interaction of factors regulating gene expression including the basal transcriptional machinery. One primary function of histone modifications in transcription regulation is serving as points of recognition for transcriptional regulators and chromatin-associated proteins [44]. 

Acetylation and methylation are the most characterized PTMs so far. Histone acetylation is controlled by two families of enzymes: histone acetyltransferases (HATs) that “write” the acetyl mark, and histone deacetylases (HDACs), that “erase” the acetyl group. On the other hand, histone methylation is catalyzed by histone methyltransferases (HMTs), whereas demethylation involves the activity of histone demethylases (HDMTs). Both epigenetic marks change quickly in response to intracellular and environmental cues and confer huge power to functional responses [43]. Unlike acetylation, which is considered an activating mark, methylation of histones can be either activating or repressive, depending on the position of the methylated residue and the extent of methylation [17,31].

### 3.1. Histone Acetylation in HSCs Activation

Little is known about the role of HAT in HSC activation. A very recent study demonstrates a strong implication of the transcriptional coactivator and HAT p300 in stiffness-mediated HSC activation [45]. p300 promotes gene transcription by acetylation not only of histones, but also transcription factors and other activators and coactivators. Stiffness induces p300 nuclear accumulation and the p300-dependent transcription of the key fibrogenic genes *α-smooth muscle actin (α-SMA*) and *connective tissue growth factor* (*CTGF*), in addition to a panel of secreted profibrogenic and tumor-promoting factors including *CXCL12, IL11, IL6, PDGFA* and *B*, and *VEGFA* [45]. Therefore, p300 may represent a novel target for suppressing liver fibrogenesis and the protumorigenic microenvironment. 

More studies have been focused on the role of HDACs in liver fibrogenesis and HSCs activation. In mammals, eighteen HDACs have been identified and, according to their structure and mechanism of action, they can be classified in four classes: HDAC1, -2, -3, and -8 belong to class I HDACs; HDAC4, -5, -6, -7, -9, and -10 are class II HDACs (with HDAC4, -5, -7, and -9 recently classified as class IIa and HDAC6 and -10 as class IIb); class III HDACs are the Sirtuin family (Sirt1-7), and HDAC11 is the only member of class IV [46]. It is clear that changes in expression of the different HDACs occur during HSC transdifferentiation, although published observations appear inconsistent. For instance, Mannaerts and colleagues [47] found that HDAC1 and 2 were downregulated at the protein level during HSC activation whereas Qin and Han did not observe changes in HDAC2 expression while HDAC1 was induced [48]. Albeit apparently contradictory, these data may be explained by temporary changes in gene expression of these HDACs, from peak induction in early-stages of activation followed by decreased levels at later stages. Similar results were found for the mRNA levels of HDAC5 and HDAC6 that present a peak of induction at day 4 of primary HSC culture and then fall until they reach the same mRNA levels as day 1 of culture [49]. Other HDACs, such as HDAC4 [48] and HDAC8 [47], seem to have a clearer progressive increase along the activation process, whereas HDAC3 presents more constant protein expression levels. However, the mRNA levels of HDAC9 and HDAC10 are significantly downregulated in HSC activation [49]. Mechanistic studies found that a class II selective HDAC inhibitor is able to partially inhibit HSCs transdifferentiation in vitro and to reduce collagen production in fibrotic livers from an in vivo model [49]. Additionally, chemical inhibition of HDAC1, HDAC2, and HDAC4 was found to correlate with apoptosis and autophagy in activated HSCs [50]. Although the mechanisms by which HDAC inhibitors induce cell death are still unknown, it was suggested that acetylation of nonhistone proteins may play a critical role [51]. 

In line with the increase in HDAC4 levels during culture-activation of HSC, a progressive decrease of global levels of N-terminal acetylation of both histones H3 and H4 has been observed [48]. Interestingly, a causal relationship between *HDAC4* elevation and matrix metalloprotease (*MMP*) gene repression was subsequently demonstrated [48]. MMPs are a family of endopeptidases highly expressed in acute injury, which are progressively repressed in fibrotic liver, favoring ECM accumulation. They are produced by quiescent HSCs and function in tissue injury and wound healing by releasing growth factors and promoting cell migration and ECM turnover [52,53,54]. When HSCs activate, they become unable to express most MMPs even under inflammatory stimulation [55]. Qin and Han ectopically expressed HDAC4 in quiescent HSCs abolishing the IL-1-mediated induction of *Mmp9* and *Mmp13* and reducing the endogenous *Mmp9* expression [48], thus supporting the implication of HDAC enzymes in gene repression during HSC transdifferentiation.

A novel link between HSC activation and histone deacetylation is the regulation of *Hepatocyte Growth Factor* (*HGF*) expression by HDAC7. In the liver HGF protects from fibrosis and inhibits HSC activation through several mechanisms. When HSCs activate, the expression of HGF is reduced, and HGF levels can be restored by treating cells with class I HDAC inhibitors or by knocking-down HDAC7 expression [56]. The transforming growth factor-β (TGFβ) pathway is another relevant modulator of liver fibrogenesis in which HDACs have also been involved. TGFβ is a key growth factor in ECM formation and remodeling. The nonspecific HDAC inhibitor trichostatin A (TSA) was used to demonstrate the implication of HDACs in controlling the TGFβ regulatory pathway in mouse C3H10T1/2 fibroblasts. TSA did not significantly affect Smad-mediated TGFβ signaling; however, ERK- and PI3K-dependent TGFβ signaling was profoundly impaired by HDAC inhibition, that of HDAC3 in particular [57]. These signaling pathways are required for the expression of a number of TGFβ-regulated genes that are also known to be suppressed by HDAC inhibitors [58,59,60]. 

Other recent work also points out a role for HDAC2, HDAC6, and HDAC8 in liver fibrogenesis through mediating the activation of TGFβ1 signaling pathway in fibrotic livers. They found that TGFβ-stimulated HSCs increased their protein levels of HDAC2, HDAC6, and HDAC8 and that chemical inhibition of these HDACs impairs TGFβ signaling and reduces liver fibrosis in a mouse model [61].

### 3.2. Histone Methylation in HSCs Activation

In addition to HDACs, HMTs and HDMTs have also been involved in the regulation of gene expression during HSC transdifferentiation. In contrast to HDACs, there is no global information about the differential expression of HMTs and HDMTs in fibrotic livers or activated HSCs. However, several studies have suggested a critical role for these enzymes in liver fibrogenesis. For instance, it was recently demonstrated that ethanol exposure promotes HSC transdifferentiation by triggering global changes in histone modifying enzymes. A significant induction in the expression of histone 3 lysine 4 (H3K4) methyltransferases, mainly that of MLL1 was observed. Upon ethanol treatment MLL1 was recruited to the proximal promoter of the elastin gene, a fibrosis-related protein, and the levels of H3K4me3 were consistently increased in this region in association with enhanced elastin expression [62]. These findings are in line with the increased levels of dimethylated (H3K4me2) and trimethylated (H3K4me3) H3K4 found in the promoters of different profibrogenic genes, such as *Col1a1/2*, upon TGFβ stimulation in mouse embryonic fibroblasts (MEFs). The deposition of these histone marks was mediated by a multi-unit protein complex known as COMPASS, which includes histone methyltransferases such as ASH2 and SET1 that accumulated on the promoter of profibrogenic genes in MEFs and HSC-T6 cells following TGFβ stimulation [63]. ASH1 is another HMT that also targets the activating mark lysine 4 on histone H3 [64]. This enzyme has been found to be highly upregulated during the transdifferentiation process of primary HSC. ASH1 directly binds to regulatory regions of the profibrogenic genes *α−SMA*, *Col1a1*, and *TIMP-1*, and TGFβ1 exerts a positive effect on their expression [65]. 

EZH2 methyltransferase is another relevant HMT involved in the development of liver fibrosis. It is the catalytic subunit of the plycomb repressive complex 2 (PCR2) and catalyzes the methylation of H3 at lysine 27 [66]. EZH2 expression is induced in HSCs when activated upon CCl_4_-mediated liver injury [67]. It was found that EZH2 expression can also be increased in HSCs when treated with TGFβ but not with PDGF-BB [68], leading to the hypothesis that EZH2 is specifically involved in TGFβ-dependent profibrotic pathways. In activated HSCs, EZH2 upregulation correlates with H3K27 methylation in exons of the Peroxisome Proliferator-Activated Receptor Gamma (*PPARγ*) gene, resulting in its transcriptional downregulation [67]. Transcriptional silencing of *PPAR*γ is crucial for enabling transdifferentiation to the myofibroblast phenotype [67,69]. PPARγ is a transcription factor involved in the maintenance of the quiescent state regulating glucose and lipid metabolism in HSCs, among many other functions. PPARγ reinforces their adipogenic phenotype, but when HSCs become activated, PPARγ expression and activity are downregulated [70,71]. 

In addition to HMTs, involvement of HDMTs activity has also been found during HSCs activation. Domain-Containing Protein 1A (JMJD1A) is a HDMT implicated in the regulation of *PPARγ* expression. In quiescent HSCs, JMJD1A reduces H3K9me2 levels on the *PPARγ* gene promoter, contributing to its expression. Consistently, siRNA-mediated JMJD1A knockdown in HSCs was found to correlate with reinforced H3K9me2 contents and the downregulation of *PPARγ* gene expression [72]. Another HDMT recently found to be involved in this is the H3 demethylase KDM4D, also known as JMJD2D. This enzyme is upregulated in activated HSCs and its knockdown in vivo inhibited fibrosis progression in the mouse CCl_4_ model [73]. Together, all these studies underscore the critical role of histone modifications in HSC transdifferentiation and the complexity of the epigenetic regulation of this process.

## 4. Crosstalk between DNA Methylation and the Histone Code: Another Layer of Complexity in Fibrogenic Activation

The overall combination of histone marks and their functional relationship with DNA methylation status, together with the presence of other regulatory factors, ultimately determine chromatin conformation and the expression level of associated genes. In other words, numerous associations exist in the recruitment of proteins and enzymatic complexes that can recognize and bind to methylated CpGs or specific histones marks or even both. Therefore, all the epigenetic events involve an intimate and complex crosstalk that include DNA methylation regulation, histones modification mechanisms, and the recruitment of transcriptional corepressors/coactivators [74,75]. 

In this context, Methyl-CpG-Binding Protein 2 (MeCP2) plays a pivotal role in the coordinated epigenetic regulation of HSC transdifferentiation. MeCp2 is a methyl-DNA-binding protein that has been demonstrated to be crucial for fibrogenesis. MeCP2-deficient mice were found to be protected from liver [67] and lung fibrosis [76], whereas MeCP2 was found to also be implicated in myofibroblast differentiation in the heart [77] and eye [78], indicating that it could be a generic core-regulator of tissue fibrosis. Mann’s group has extensively studied the induction of expression of the *MeCP2* gene during HSCs activation, that along with other proteins constitutes a nuclear complex able to repress transcription of target genes by inducing either DNA methylation or the deposition of silencing histone marks [33,65,67]. MeCP2 is able to repress transcription from methylated gene promoters. It was demonstrated that this is a mechanism responsible for the epigenetic silencing of the key target gene *PPARγ* in HSCs. MeCP2 binds to the *PPARγ* promoter where it facilitates methylation of lysine 9 of histone 3 (H3K9), that in turn provides binding sites for Heterochromatin Protein 1 (HP1) and promotes transcriptional silencing. Moreover, MeCP2 stimulates expression of *EZH2*, which is recruited to the downstream coding region of *PPARγ* where it increases the methylation levels of H3K27 mediating polycomb-regulated transcription silencing [67]. Thus, MeCP2 promotes *PPARγ* silencing by two mechanisms, inhibiting both the initiation of transcription and elongation of transcription. In addition to its repressive role, and as mentioned before regarding EZH2 expression, MeCP2 has been reported to also promote transcriptional activation [79,80]. MeCP2 is able to mediate the induction of expression of positive regulators of profibrogenic genes as is the case of the histone lysine methyltransferase ASH1 [65]. These MeCP2-mediated mechanisms are one example of coordinated activity between DNA methylation and histones modifications, reflecting the relevance of the crosstalk between epigenetic events, but also evidencing the paradoxical function of epigenetic modifiers that can work as coactivators and corepressors depending on their specific interactions and molecular context. The mechanisms of this dual activity are not fully understood, but interestingly a study in neural cells suggested that binding of MeCP2 to 5-hydroxymethyl-cytosine residues in CpGs, instead of 5-methyl-cytosines, enables MeCP2 to function as a stimulator of transcription [80]. 

In addition to MeCP2, other epigenetic complexes involving different crosstalk are being elucidated in the context of HSC transdifferentiation. We recently observed that the epigenetic complex formed by the H3K9 methyltranferase G9a and the DNA methyltransferase DNMT1 seems to play a relevant role in the TGFβ-mediated responses of HSCs [81]. Dual inhibition of G9a and DNMT1 with a new class of drugs was able to inhibit the growth and activation of human HSCs and impaired the induction of profibrogenic genes in response to TGFβ. Moreover, ablation of G9a and DNMT1 activities also resulted in the reinduction of the expression of *PPARγ*, which as previously mentioned is a master regulator of HSC quiescence and differentiated status. We found that G9a and DNMT1 inhibition reduces DNA and H3K9 methylation levels in the promoter region of *PPARγ*, indicating that both G9a and DNMT1 might cooperate in the repression of this gene when HSCs activate.

## 5. Translational Perspectives

### 5.1. Epigenetic Marks as Biomarkers for Liver Fibrosis

Epigenetic signatures have been considered for their potential application as prognostic and diagnostic biomarkers. In this field, genomic DNA methylation patterns in human liver can distinguish patients with different stages of nonalcoholic fatty liver disease (NAFLD), alcoholic liver disease (ALD), and liver fibrosis [82,83,84]. Zeybel and colleagues investigated whether DNA methylation profiles could distinguish patients with mild from those with advanced/severe fibrosis in NAFLD and ALD [82]. They found that DNA methylation status at specific CpGs might be useful for predicting progression of the disease and development of fibrosis, further supporting their previous studies on DNA methylome signatures and NAFLD patient’s stratification [83]. Potential limitations of this approach relied on analysis of tissue provided by liver biopsy, an invasive procedure with associated risks that may complicate the translation of these findings to clinical practice. 

Another promising strategy is based on the analysis of DNA methylation in circulating cell-free DNA isolated from patient’s plasma, and therefore would be a type of “liquid biopsy” [84,85]. This method has the potential to stratify fibrosis grade with high accuracy and was based in the changes observed in DNA methylation density at the *PPARγ* promoter in circulating cell-free DNA. Moreover, quantification of this epigenetic mark provided a powerful stratification tool, identifying NASH patients that had progressed to severe fibrosis.

### 5.2. Epigenetic Drugs Targeting the Fibrotic Stroma

The epigenetic events described above that modulate HSC transdifferentiation and activation are emerging as novel and attractive therapeutic targets to treat hepatic fibrosis. Many epigenetic drugs are currently being tested as anticancer agents in clinical trials, and several of them are already used in the clinic. DNMTs inhibitors such as 5-azacytidine, decitabine, and more recently guadecitabine, are being tested in a variety of neoplasias, including hematological malignancies [86,87] and solid tumors like HCC [88,89]. These drugs are apparently well tolerated, suggesting that they might be also useful to treat or prevent liver fibrogenesis. 

HDACs and HMTs could also constitute potential targets in liver fibrosis. Small molecules targeting histone modifying enzymes, particularly inhibitors of HDACs, are an important component of the epigenetic drug market. They are currently licensed for the treatment of different malignancies and are being explored as therapies for many other tumors [90]. Pharmacological HDAC inhibitors present antiproliferative and proapoptotic properties in HSCs [50,91,92]. They are able to reverse myofibroblast differentiation and display antifibrotic effects in different organs and tissues such as lung, kidney, and skin [93,94,95,96]. 

In cultured human HSCs, sodium valproate, a broad class I and II HDAC inhibitor also used in the clinic to treat a variety of neurological disorders, demonstrated antifibrotic properties by inhibiting the expression of *Col1a1* and *TGFβ1* without causing toxic effects [97]. Valproate was also capable of blocking myofibroblast differentiation and fibrogenesis in mouse models of liver fibrosis [47]. Very recently, another HDAC inhibitor, suberoylanilide hydroxamic acid (SAHA or Vorinostat), was also able to suppress TGFβ-signaling and demonstrated antifibrotic effects in an animal model of CCl_4_-induced liver fibrosis [61]. The HDAC inhibitor TSA has also emerged as a potent HSC activation inhibitor. TSA treatment of cultured-activated HSCs resulted in attenuated proliferation and inhibition of *α−SMA* and *Col1a1* and *Col1a2* genes transcription [59,98,99,100]. TSA also demonstrated an antifibrotic effect in the in vivo model of CCl_4_-induced hepatic fibrosis, improving liver function and reducing ECM accumulation [100]. Although HDAC inhibitors show promising antifibrotic effects in vitro and in vivo, their mechanisms of action are still elusive. Further understanding of the specific roles of each HDAC in fibrogenesis may lead to the application of selective inhibitors instead of general HDAC inhibitors, therefore reducing unwanted side effects.

Specific inhibitors of relevant HMTs, such as EZH2, also present antifibrotic properties. The EZH2 inhibitor 3-Deazaneplanocin A (DZNep) blocks HSC activation [33] and attenuates fibrosis in vivo [68,101]. Targeted delivery of the drug to HSCs in mice livers was achieved by DZNep encapsulation in liposomes coated with C1-3 ScAb, an HSC-specific antibody. These drug-loaded liposomes suppressed the progression of pre-established liver fibrosis induced by CCl_4_ treatment in mice [101]. As previously mentioned, novel molecules with dual inhibitory capacity simultaneously targeting DNMT1 and the HMT G9a have demonstrated efficacy in reducing HSCs proliferation and impairing TGFβ mediated activation [81]. Preclinical evidence indicates that these dual inhibitors are well tolerated and do not show signs of toxicity when administered in vivo for long periods of time [102].

## 6. Conclusions

This review aims to highlight relevant epigenetic mechanisms involved in the activation of HSCs and liver fibrogenesis. As summarized in Figure 1A, liver injury triggers global epigenetic remodeling in fibrotic livers and activated HSCs. Accumulating experimental evidence indicates that these events control HSC activation and progression of fibrosis. A list of differentially expressed genes regulated by epigenetic mechanisms and discussed in this review is given in Table 1. Although most of the studies we summarized in this review are either focused on DNA methylation patterns and on the characterization or PTMs of histones alterations (Table 2), it is very important to emphasize the role of interactions between the different epigenetic events. An example of coordinated epigenetic regulation of gene transcription discussed in this review is represented in Figure 1B, reflecting the complexity of epigenetic crosstalk. 

The current challenge is to identify the key relevant epigenetic events or epigenetic modifiers driving liver fibrogenesis. This would allow not only the delimitation of potential therapeutic targets, but also the repositioning of epigenetic drugs that are currently used for the treatment of neoplastic conditions, but that may harbor antifibrotic properties. We have also discussed the potential utility of epigenetic marks as biomarkers of liver disease. Although this field has been minimally studied, its clinical application can be crucial to improve the diagnosis and prognosis of patients with CLD.

## Figures and Tables

**Figure 1 ijms-20-02507-f001:**
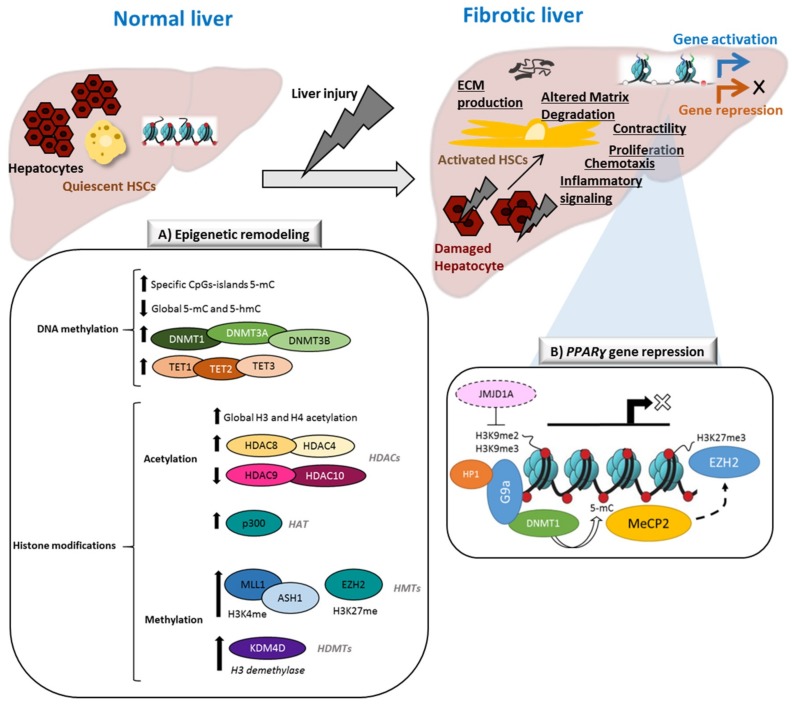
Schematic representation of epigenetic remodeling events in HSCs in the context of liver fibrosis. Liver injury initiates the transdifferentiation of quiescent hepatic stellate cells (HSCs) to their activated phenotype. Specific gene expression/repression processes respond to specific phenotypic changes, including proliferation, contractility, EMC remodeling, chemotaxis, or inflammatory signaling. (**A**) Global epigenetic changes in activated HSCs, including DNA methylation patterns, DNA-methylation modifiers (DNMTs and TETs), histone post-translational modifications (PTMs) patterns, histone acetyl transferases (HATs), histone deacetyltransferases (HDACs), histone methyltransferases (HMT), and histone demethylases (HDMTs). (**B**) Example of coordinated epigenetic regulation of *PPARɣ* transcriptional repression in activated HSCs.

**Table 1 ijms-20-02507-t001:** Genes epigenetically regulated in activated HSCs.

Expression	Specific Genes	Regulatory Mechanism
Increased expression	*Apc2* [35]	Promoter-DNA hypomethylation
	*Wnt5a* [39]	Promoter-DNA hypomethylation
	*Actg2* [36]	Promoter-DNA hypomethylation
	*Loxl1* [36]	Promoter-DNA hypomethylation
	*Loxl2* [36]	Promoter-DNA hypomethylation
	*Col4a1/2* [36]	Promoter-DNA hypomethylation
	*α-SMA* [45,65]	p300-dependent transcription
	*Elastin* [62]	MLL1-mediated H3K4 methylation
	*Col1a1/2* [64,65]	COMPASS-mediated H3K4 methylation
	*TIMP-1* [65]	ASH1-mediated H3K4 methylation
		ASH1-mediated H3K4 methylation
	*TGFβ1* [65]	ASH1-mediated H3K4 methylation
Decreased expression	*Adamts9* [36]	Promoter-DNA hypermethylation
	*Mmp15* [36]	Promoter-DNA hypermethylation
	*Smad7* [40]	Promoter-DNA hypermethylation
	*Pten* [41]	Promoter-DNA hypermethylation
	*Mmp9* [48]	HDAC4-mediated histone deacetylation
	*HGF* [56]	HDAC7-mediated histone deacteylation
	*PPARγ* [33,65,67]	Promoter DNA-hypermethylation
		EZH2-mediated H3K27 methylation
		G9a-mediated H3K9 methylation
		MeCp2-mediated repressive complex recruitment

**Table 2 ijms-20-02507-t002:** Epigenetic factors activated or suppressed in activated HSCs and its target genes.

Activated	Inactivated	Associated Epigenetic Marks
DNMTs [35]	TETs [42]	Global DNA hypomethylation Promoter-specific hypermethylation
p300 [45]		Histone and nonhistone targets acetylation
HDAC1* [48]; HDAC4 [48]; HDAC5* [49]; HDAC6* [49]; HDAC8 [47]	HDAC9 [49]; HDAC10 [49]	Changes in H3 and H4 acetylation patterns
MLL1 [62]; ASH2 [63]; SET1 [63]; ASH1 [64]		Increased gene-specific H3K4me2/3
EZH2 [67,68,69]		Increased gene-specific H3K27me3
G9a [81]		Increased gene-specific H3K9me2
	JMJD1A [72]	Reduced gene-specific H3K4me2/3
JMJD2D [73]		H3K9 demethylation

*At early stages of HSC activation.

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
