# Peer review of "Epigenetic Mechanisms in Hepatic Stellate Cell Activation During Liver Fibrosis and Carcinogenesis"

_ijms, 2019, doi:10.3390/ijms20102507_

Reviewer 1 Report

In this manuscript, the authors summarized the epigenetic alterations during the course of hepatic stellate cells (HSCs) activation. They also provide detailed explanations about general mechanisms of epigenetic regulation, which is helpful for researchers who are not familiar with epigenetics. Furthermore, of interest, its applications for liver fibrosis, as a diagnostic and predictive marker, and a therapeutic target, are described. This review article is well written, and attractive for clinical and basic researchers of liver diseases. However, the following points should be considered.

1. The title does not match with the contents. Although HCC is a consequence of liver fibrosis, epigenetic changes in HSCs, which bring about HCC, were not described. In addition, the title is confusing, and may mislead some readers expecting epigenetics in cancer associated fibroblasts.

2. The number of figures is too few. The following tables are useful for readers:

A) a table summarizing genes, which are reported to be epigenetically regulated in HSCs, and its regulatory mechanisms (ex. CpG methylation, H3K4me3...)

B) a table summarizing epigenetic factors activated or suppressed in transdifferentiated HSCs, and its target genes

3. The text size in Figure 1 is too small to read. This should be revised.

Author Response

Response to Reviewer 1 Comments

Point 1. The title does not match with the contents. Although HCC is a consequence of liver fibrosis, epigenetic changes in HSCs, which bring about HCC, were not described. In addition, the title is confusing, and may mislead some readers expecting epigenetics in cancer associated fibroblasts.

Thank you very much for your consideration. Indeed, we have changed the title to: Epigenetic mechanism in Hepatic Stellate Cells Activation during Liver Fibrosis and Carcinogenesis.

Point 2. The number of figures is too few. The following tables are useful for readers:

A) a table summarizing genes, which are reported to be epigenetically regulated in HSCs, and its regulatory mechanisms (ex. CpG methylation, H3K4me3...)

B) a table summarizing epigenetic factors activated or suppressed in transdifferentiated HSCs, and its target genes

We have included Table 1 and Table2 according to the reviewer suggestions summarizing both, genes and factors implicated in the activation of Hepatic Stellate Cells.

Point 3. The text size in Figure 1 is too small to read. This should be revised.

 We have enlarged the text size in Figure 1.

 Beside this, moderate English changes have been performed in the manuscript following  the reviewer recommendation.

Reviewer 2 Report

The review by Marina Barcena-Varela et al. is comprehensive and timely. Reviewing epigenetic changes and signaling in HCC is important to the field. I have no major concerns except some spell checks and in figure 1 Normal quiescent liver should labelled as Normal liver.

Author Response

Response to Reviewer 2 Comments

The review by Marina Barcena-Varela et al. is comprehensive and timely. Reviewing epigenetic changes and signaling in HCC is important to the field. I have no major concerns except some spell checks and in figure 1 Normal quiescent liver should labelled as Normal liver.

In Figure 1, Normal quiescent liver has been replaced with Normal Liver.

Round  2

Reviewer 1 Report

The authors had adequately addressed my comments.

I have no further comments.